# Reconstructing the incidence rate and immune fraction of the population via a single snapshot survey: A case study of COVID-19 in Japan

Yuta Okada[1], Hiroshi Nishiura[1,2]*

1 Kyoto University School of Public Health, Kyoto, Japan, 2 Center for Health Security, Graduate School of Medicine, Kyoto University, Kyoto, Japan

* nishiura.hiroshi.5r@kyoto-u.ac.jp

## Abstract

While the global health burden of COVID-19 continues, multifaceted epidemiological surveillance is required to monitor the epidemic's dynamics and its population-wide risk. By collecting information that is used in conventional vaccine effectiveness studies through questionnaire surveys, we proposed a simple framework using a population-wide snapshot questionnaire survey to estimate the incidence and protective effect of immunity by natural infection or vaccination against the SARS-CoV-2 JN.1 variant. Our results revealed that in Japan in February 2024, the personal risk of diagnosed infection was substantially higher in younger adults and risk was heterogenous across prefectures. Diabetes mellitus (relative hazard ratio 1.8; 95% credible interval [CrI] 1.1, 2.9), neoplastic disorders (5.2; 95% CrI 3.1, 8.6), immunological suppression (2.6; 95% CrI 1.3, 4.6), respiratory diseases (2.2; 95% CrI 1.4, 3.3), and cardiovascular disease (2.3; 95% CrI 1.3, 3.9) were risk factors for diagnosed infection. The highest peak protection after infection was after exposure to pre-XBB.1.5 Omicron variants (52.0%; 95% CrI 33.2, 68.7), whereas the XBB.1.5 monovalent vaccine provided the highest protection (45.1%; 95% CrI 37.8, 52.7) among three vaccine types. Notably, the peak protection of the bivalent Wuhan + Omicron BA.1/5 vaccine was substantially lower than other vaccines (28.7; 95% CrI 17.3, 40.6). By statistically matching the respondent cohort to the 2020 population census, we revealed that the national COVID-19 incidence rate in February 2024 by age group was highest (4.73%; 95% CrI 4.17, 5.38) and lowest (1.19%; 95% CrI 0.94, 1.47) among those aged 20–29 years and 60–69 years, respectively. The force of infection measured by diagnosed infection was high and more heterogeneous in younger groups, whereas younger populations were more concentrated than older populations in low-protection regions. Our framework revealed biological and epidemiological insights into protection and risk of diagnosed infection from past immunizing events and personal attributes during the JN.1-dominant period. Moreover, we proposed a framework for the rapid evaluation of epidemiological dynamics whose application is not limited to COVID-19.

**Data availability statement:** The anonymized data from the questionnaire survey in this study as well as data from the 2020 census that are used in this study are provided as supplementary data. The codes used for the analyses in this study are also accessible at https://github.com/pk2393/covid_qs_feb2024. (Archive on Zenodo: https://doi.org/10.5281/zenodo.17221098).

**Funding:** This work was supported by the SECOM Science and Technology Foundation (YO), Health and Labour Sciences Research Grants (grant numbers 20CA2024, 21HB1002, 21HA2016, and 23HA2005 to HN), the Japan Agency for Medical Research and Development (grant numbers JP24fk0108710 and JP24fk0108685 to HN), JSPS KAKENHI (grant numbers 21H03198 and 22K19670 to HN), the Environment Research and Technology Development Fund (grant number JPMEERF20S11804 to HN) of the Environmental Restoration and Conservation Agency of Japan, the Daikin GAP Fund of Kyoto University (HN), HU-RIZONT international research excellence program (Rapid-GRIP project: 2024-1.2.3-HU-RIZONT-2024-00034 to HN); the World Health Organization (HN), the Japan Science and Technology Agency SICORP program (grant numbers JPMJSC20U3 and JPMJSC2105 to HN), the CREST program (grant number JPMJCR24Q3 to HN), and the RISTEX program for Science, Technology, and Innovation Policy (grant number JPMJRS22B4 to HN). The funders had no role in the study design, data collection and analysis, decision to publish, or preparation of the manuscript.

**Competing interests:** The authors have declared that no competing interests exist.

## Author summary

The questionnaire survey is a powerful tool for generating population-wide data in fields including public health, but its application in tracking the immunological characteristics and incidence rate of COVID-19 is limited. We proposed a new framework to leverage the advantages of this tool by combining an Internet-based questionnaire survey with statistical models. Two findings emerged from our study, which focused on self-reported diagnosis of COVID-19 infection in February 2024, when the JN.1 variants of severe acute respiratory syndrome coronavirus 2 were dominant in Japan. First, the overall reduction in the relative hazard of infection from vaccinations or natural infection was at most roughly 60% and had substantial time decay. Second, compared with older age groups, a higher incidence rate and lower protection from acquired immunity were observed in young age groups, with substantial heterogeneity within each age-stratified population. This study revealed important immune profiles against the emerging JN.1 variant and identified an effective framework that can be added to the array of epidemiological programs that complement routine public health surveillance.

## 1. Introduction

The coronavirus disease 2019 (COVID-19) pandemic has caused extensive mortality since its emergence in 2019 in Wuhan, China [1–3]. Although COVID-19 continues to have a substantial impact on health [4–9], epidemiological surveillance efforts have been globally downgraded to the extent that tracking the absolute number of infections and transmission dynamics at the national and regional levels has become impossible in most countries. Furthermore, at more than 4 years since the emergence of the virus, conducting studies on the effectiveness of vaccine- or infection-induced immunity has become increasingly complex given the growing complexity of the individual-level history and antigenic details of vaccination and infection and the very small size of the population naïve to both infection and vaccination. In addition, adherence by individuals to precautionary measures has varied considerably over time given society's increasingly diverse views toward COVID-19. Because of the increased number of infections and the accelerated evolution of the virus, the rapid turnover of SARS-CoV-2 variants has also thwarted conventional vaccine studies that quantify protection level by measuring humoral immune response. Given that the results of such studies reach the public domain in a few months, a new SARS-CoV-2 variant with a substantial transmission advantage or immune evasion capability can begin causing another large epidemic. To facilitate the use of population immune profiles as part of real-time epidemiological risk assessment, approaches that expand the coverage and timeliness of epidemiological surveillance are urgently needed.

Efforts to enhance the coverage and timeliness of epidemiological surveillance beyond routine public health monitoring have been made before the emergence of

COVID-19. During the COVID-19 pandemic, most countries ran universal or sentinel surveillance to monitor the virus, whereas various complementary approaches such as wastewater surveillance, digital syndromic surveillance, and participatory surveillance, including population-wide questionnaire surveys, increasingly proved useful during the pandemic [10–19]. These approaches provide tools to capture the epidemiological trend of the epidemic. However, to capture the entire scale of an epidemic, participatory or questionnaire-based survey can also be practical options. Cross-sectionally, a participatory active survey involving biological samples has been proven valuable [19], although financial and labor costs may limit its deployment. Additionally, although pure questionnaire surveys cannot collect biological evidence, they can 1) play an important role in quantifying transmission dynamics, 2) be implemented at a low cost, and 3) be conducted rapidly. Published studies have deployed questionnaire surveys together with epidemiological indicators such as the case-fatality ratio to estimate the population-wide incidence of influenza in the United States [18,20]. These studies were based on telephone surveys; Internet-based surveys can further enhance the effectiveness of questionnaire surveys in the context of epidemic risk assessment.

Studies in the earlier COVID-19 pandemic period that estimated the immunity effects of infection or vaccination after adjustment for individual factors, including health, biological, and social status, have been extremely successful [21,22]. However, because the variability of personal precautionary behaviors is currently high, regularly updating knowledge on the effect of factors that modulate the susceptibility to infection is desirable. Furthermore, providing national or sub-population-level "susceptibility mapping" by attributes such as age can be invaluable for guiding public health decisions that optimize non-pharmaceutical interventions or vaccination strategies for epidemic control [23]. Published studies have provided useful frameworks for susceptibility mapping; however, these frameworks do not apply to the current global situation in which population-level data on case counts and vaccination records are no longer available [24,25].

In Japan, the universal COVID-19 case count was officially stopped in May 2023, and national vaccination records (e.g., as a registration system) are also currently unavailable. Some projects are underway to recover, at least partially, case counts that are equivalent to universal case counts before May 2023, and a government-led project is underway to conduct seroepidemiological surveys among voluntary participants and collect residual serum samples from healthcare facilities and blood donors [10,26,27]. However, although these efforts are very useful for addressing the current lack of comprehensive epidemiological data, the coverage of these projects is insufficient for rapidly elucidating population-level incidence and immunity. Moreover, at the time of writing, vaccine effectiveness in Japan against infection or severe disease from COVID-19 had only been evaluated up to the bivalent mRNA vaccine and for COVID-19 variants that preceded the JN.1 variants [28–33]. Thus, identifying a framework that enables timely updates of personal-level immunity against the latest emerging SARS-CoV-2 variants is another important goal. In this context, we propose a web-based questionnaire approach to estimate not only personal-level immunity and risk-modifying attributes, including health conditions, but also the population-level incidence of COVID-19 infection and immune protection.

Motivated by the issues described above, we proposed a practical framework that uses an Internet-based social survey to estimate the COVID-19 incidence rate and immune protection against the virus from past infection or vaccination. Using survey respondents' personal background information, we scaled up the survey results to reconstruct the population-wide incidence and immune profiles. This process was achievable only via a single snapshot survey.

## 2. Materials and methods

### 2.1 Ethics statement

The questionnaire survey conducted in this study was approved by the Ethics Committee of the Graduate School of Medicine, Kyoto University (number R4232). Written informed consent was obtained from all participants of the survey by ticking "agree" square of the web-survey page before enrollment. No ethical approval was required for the census data given its openly accessible nature.

## 2.2. Data used in the study

**2.2.1. Questionnaire survey.** The three key objectives of this study were to 1) estimate the incidence rate of COVID-19 in February 2024 in Japan, 2) characterize the effectiveness of vaccine or infection-induced immunity (peak immunity, decay over time, difference by history of vaccination or infection), and 3) estimate the impact of individual-level attributes including health status on the risk of contracting COVID-19.

Given these objectives and by referring to previously reported information on risk factors for infection and severe outcomes of COVID-19 [34–38], we designed the questionnaire to collect the following information from all respondents:

1. Diagnosis of COVID-19 between 1 and 29 February 2024

2. Biological and health-related background: age, sex, and underlying chronic conditions

3. Social background: profession (occupation), educational status, and household size

4. History of COVID-19 vaccination and natural infection.

The diagnosis of COVID-19 was defined as meeting at least one of the following two criteria, regardless of the presence or absence of symptoms.

• A positive result from a PCR test. The testing location (e.g., hospital, public health center, testing center) and sample type (e.g., saliva, nasal swab) are not specified.

• A positive result from an antigen test (test kit).

Note that, because testing to capture asymptomatic COVID-19 cases were uncommon since 2022 in Japan, the protection against diagnosed COVID-19 infection as discussed in the following section can be deemed as essentially comparable to "symptomatic COVID-19 infection". Further details of each questionnaire item can be found in the S1 Text.

An Internet-based questionnaire survey was conducted by a private company (MelLinks Co. Ltd, Tokyo, Japan) in Japan that specializes in Internet-based social surveys using a method that was used in a published study [39]. Among the group of pre-registered monitor respondents that are retained by MelLinks, the recruitment of respondents started on 7 March 2024 and ended on 13 March 2024. The invitation to participate remained open until the prefectural distributions of respondents by age were proportional to the census data.

The questionnaire was designed to collect information on a monthly time scale, and therefore we excluded 1) those who had a history of both infection and vaccination in February 2024 and 2) those who reported infection in January 2024, to exclude the potential overlap of the illness period between January and February 2024. The key descriptive features are summarized in S1 Table, S2 Table and S1 Fig, and the results are in S1 Data.

**2.2|2. Census data of Japan.** To perform the statistical weighting of survey responses, we retrieved the demographic data from the 2020 Population Census of Japan conducted by the Statistics Bureau of the Ministry of Internal Affairs and Communications [40]. We used the following data categories that are stratified by sex, age group, and prefecture (S2-S6 Data):

• Male and female population aged 20 years or older at the national level

• Age distribution by prefecture and sex

• Job category distribution by age and sex in each prefecture

**2.2.3. Data on SARS-CoV-2 variants and vaccine types in Japan.** We retrieved the historical data on the proportion of SARS-CoV-2 variants in Japan from the National Institute of Infectious Diseases website [41]. We classified the period between January 2020 and January 2024 into the following three antigenically distinct periods for subsequent analyses:

- Pre-Omicron period: January 2020–December 2021

- Pre-XBB Omicron period: January 2022–April 2023

- XBB period: May 2023–December 2023

Notably, SARS-CoV-2 JN.1 variants, the descendants of BA.2.86, were the dominant strains in February 2024. The types of mRNA vaccines in Japan were classified by period as follows [42–44]:

- Monovalent vaccine adapted to ancestral (Wuhan) strain: February 2021–September 2022

- Bivalent vaccine adapted to ancestral (Wuhan) & Omicron BA. 1/5: October 2022–September 2023

-  Monovalent vaccine adapted to XBB.1.5: September 2023–January 2024

For most periods, vaccines manufactured by Pfizer Inc. and Moderna Inc. were available and publicly administered free of charge in Japan for those aged 12 years or older. However, we did not discriminate between these vaccine types to ensure simplicity and because of the practical difficulty in collecting details on vaccine type from all of the respondents.

**2.2.4. External epidemiological data on COVID-19 in Japan.** Though there is no gold-standard epidemiological data on COVID-19 in Japan that represents the nationwide incidence rate as of February 2024, we retrieved two types of datasets from external sources to compare with our estimates. The first is the serological survey conducted by public health agencies in Japan using either residual serum samples from commercial clinical testing laboratories or residual blood samples from donated blood testing. [45–48] Surveys on serum samples from commercial laboratories during January 6th – 17th and March 2nd – 12th (anti-N, anti-S antibody) and surveys on donated blood samples during January 9th -23rd and March 4th – 18th (anti-N antibody) were retrieved. The second is the national case count estimates provided by a website supported by Moderna, Inc. We retrieved their national case count estimates (only point estimates are provided) on a daily basis from February 1st – 29th for age group 20–59 and age 60 or over. (S7 Data) [27] The description of the underlying data and method can be found in Miyawaki et al., where it is described that the data consists of clinical practice information from approximately 4,200, or about 4% of all primary care physician clinics in Japan. [26] Note that both serological surveys and the Moderna case count estimates only provide estimates of seroprevalence or case counts and do not include information on individual-level health conditions and personal attributes.

## 2.3. Statistical model and inference in the study

**2.3.1. Modeling immune decay after vaccination or infection before january 2024.** In analogy to how vaccine effectiveness in an epidemiological sense is generally expressed as $1 - HR$ ($HR$: hazard ratio) or $1 - RR$ ($RR$: relative risk), we hereafter use the letter $V$ to represent the effectiveness of protection conferred by vaccination or infection. We modeled $V_i$, the protection level against infection of an individual $i$, as a function of time since the last immunizing event, which may occur either after vaccination or infection. For those without a history of infection or vaccination, we assumed $V_i = 0$. Otherwise, we assumed a biexponential waning of protection analogous to the antibody-waning model of Hogan et al, which is based on the biological process of fast and slow waning immunity against symptomatic SARS-CoV-2 infection suggested in Khoury et al. [49,50] In contrast to Hogan et al., we directly applied the biexponential model to the protection to keep the model simple yet continue to reflect the biological dynamics of antibodies that offer protection. The model for protection, or, the relative hazard reduction at calendar time $t$ was as follows:

$$V_i(t) = v_{E_i} \left[ f_{E_i} \exp\left(-\gamma^{\{1\}}\left(t - \tau_i\right)\right) + \left(1 - f_{E_i}\right) \exp\left(-\gamma^{\{2\}}\left(t - \tau_i\right)\right) \right],$$

(1)

where $E_i$ stands for the last immunizing event of an individual $i$, $v_{E_i}$ is the peak protection level for $E_i$, and $\tau_i$ is the calendar time of the last immunizing event for individual $i$. Both $\gamma^{\{1\}}$ and $\gamma^{\{2\}}$ are exponential decay rates that characterize

short- and long-lasting protections, and $f_{E_i}$ is the fraction of the contribution of short- and long-lasting protections at the peak just after immunization. We assumed that $f_{E_i}$ only differs between infection and vaccination, and $\gamma^{\{1\}}$ and $\gamma^{\{2\}}$ are fixed parameters regardless of $E_i$. Note that in Eq. 1, $\gamma^{\{1\}}$ and $\gamma^{\{2\}}$ are shared across exposure type, and $f_{E_i}$ is shred across exposures belonging to either infection or vaccinations, whereas only $v_{E_i}$ is exposure-specific.

**2.3.2. Modeling of the risk of infection by immunization status and other individual-level factors.** First, we assumed that the susceptibility of an individual $i$ is purely attributable to a specific (acquired) immunity that can be written as

$$s_i(t) = 1 - V_i(t). \tag{2}$$

Then, we modeled the FoI for an individual $i$ as follows:

$$\Lambda_i(t) = \exp\left(\sum_k \beta_k x_{k,\,i} + \delta_{pref_i}(t)\right), \tag{3}$$

where a linear combination of covariates $x_{k,\,i}$ is adjusted by prefectural random effects. Here, note that the force of infection refers to the hazard of self-reported diagnosed infection in the present study. The first six terms of $\sum_k \beta_k x_{k,\,i}$ including the intercept represent the baseline hazard for each age group as

$$\exp\left(\beta_0 + \beta_1 x_{1,\,i} + \beta_2 x_{2,\,i} + \beta_3 x_{3,\,i} + \beta_4 x_{4,\,i} + \beta_5 x_{5,\,i}\right), \tag{4}$$

where $\exp\left(\beta_0\right)$ is modeled as the baseline hazard of 20–29 years, and $x_1 \sim x_5$ are dummy variables that represent the age group to which individual $i$ belong. The full list of covariates is provided in Table 1.

Upon exposure to $\Lambda_i(t)$, individual $i$ experiences the effective hazard $s_i(t)\Lambda_i(t)$, considering the effect of protective immunity. Thus, the probability of individual $i$ being infected from time $t_1$ to $t_2$ can be expressed as

$$p_i\left(t_1,\ t_2\right) = 1 - \exp\left(-\int_{t_1}^{t_2} s_i(t)\Lambda_i(t)dt\right). \tag{5}$$

In this study, all of the information from respondents in our dataset was chronologically discretized by month. Moreover, our focus was the occurrence of infection in February 2024. Therefore, assuming that $s_i$ and $\Lambda_i(t)$ were constant throughout February 2024 (i.e., 1 month$=t_2$-$t_1$), we simplified the notation of $p_i\left(t_1,\ t_2\right)$ as

$$p_i = 1 - \exp\left(-s_i\Lambda_i\right), \tag{6}$$

where $s_i$ and $\Lambda_i$ represent constant susceptibility and FoI, respectively, for individual $i$ throughout February 2024. Using $p_i$, the binomial likelihood for observing the questionnaire-based incidence data $D^Q$ is expressed as follows:

$$L_1\left(D^Q|\Theta\right) = \prod_{i=1}^N p_i^{y_i}\left(1 - p_i\right)^{1-y_i}, \tag{7}$$

where $y_i = 1$ (or 0) indicates that individual $i$ was diagnosed (or not diagnosed) of COVID-19 in February 2024 and $\Theta = \{v,\ \gamma,\ \beta,\ \delta\}$.

**2.3.3. Estimation of population-level incidence and immunity.** To estimate the population-wide incidence rate and immune profile in the entire Japanese population, the statistical weighting of the results from individuals who responded

**Table 1. Estimates of diagnosis-based FoI and relative hazard ratios by personal attribute.**

| Variable | Estimated values (95% CrI) |
|---|---|
| Intercept | 0.017 (0.011, 0.025) |
| age group 30–39 (years) | 1.040 (0.766, 1.447) |
| age group 40–49 (years) | 0.768 (0.528, 1.092) |
| age group 50–59 (years) | 0.556 (0.352, 0.867) |
| age group 60–69 (years) | 0.372 (0.214, 0.609) |
| age group 70 years and older | 0.367 (0.204, 0.608) |
| Female Sex (dichotomous) | 0.851 (0.657, 1.084) |
| Diabetes Mellitus | 1.832 (1.107, 2.949) |
| Neoplastic Disorders | 5.249 (3.130, 8.563) |
| Immunologically Suppressed | 2.599 (1.337, 4.558) |
| Respiratory Diseases | 2.174 (1.361, 3.342) |
| Cardiovascular Diseases | 2.279 (1.262, 3.910) |
| Cerebrovascular Diseases | 1.503 (0.756, 3.305) |
| Liver Disorders | 1.911 (0.957, 3.717) |
| Obesity (BMI>30kg/m$^2$) | 0.732 (0.339, 1.273) |
| Smoking habit | 1.306 (0.950, 1.797) |
| Alcohol drinking habit | 0.900 (0.589, 1.276) |
| Household size $\geq 2$ | 0.847 (0.631, 1.127) |
| History of Previous Infection | 7.687 (5.608, 10.70) |

to our survey was needed. The Japanese 2020 census was used as the reference data for this weighting. The following describes the statistical weighting framework, a conceptual variant of the Raking method, which is used to match survey data from a sampled group to a reference population [51]:

Let $D_c = (q_{c, 1}, q_{c, 2}, \ldots, q_{c, M_{D_c}})$ be arbitrary population-wide data, where $c$ represents the data category of $D_c$, $M_{D_c}$ is the number of subcategories in $D_c$, and $q_l$ represents the number of people belonging to subcategory $l \in \{1, 2, \ldots, M_{D_c}\}$.

Given that the subset of questionnaire-based data $D_c^Q = (q_{c, 1}^Q, q_{c, 2}^Q, \ldots, q_{c, M_{D_c}}^Q)$ that also represents data category $c$, consider some adjustment or weighting to $D_c^Q$ to obtain a dataset comparable to $D_c$. For this aim, we introduced a weighting $\alpha = [\alpha_1, \alpha_2, \ldots, \alpha_N]$, $\sum \alpha_j = 1$, where each $\alpha_j$ is a weight for each individual $j$ who responded to the questionnaire. Then, finding the optimal weight $\alpha$ is essentially inferring $\alpha$ based on the likelihood $L(D_c|\pi_{D_c|\alpha})$ defined by the model

$$D_c \sim Multinomial \left( \sum_{l=1}^{M} q_{c, l}, \ \pi_{D_c|\alpha} \right), \tag{8}$$

$$\pi_{D_c|\alpha} = \left[ \pi_{c, 1|\alpha}, \ \pi_{c, 2|\alpha}, \ \ldots, \ \pi_{c, M_{D_c}|\alpha} \right], \tag{9}$$

$$\pi_{c, l|\alpha} = \frac{\sum_{j \in l_{D_c}} \alpha_j}{\sum_{l_{D_c}} \sum_{j \in l_{D_c}} \alpha_j}, \tag{10}$$

where $\sum_{j \in l_D} \alpha_j$ represents the sum of $\alpha_j$ over all $j$ that belong to category $l_D$.

In this study, the three data categories from the 2020 census described above were used for the above-mentioned fitting protocol. By naming the set of these three data categories as $C_{census}$, the likelihood of observing $D$ given $\alpha$ can be written as a combination of multinomial likelihoods:

$$L_2\left(D|\alpha\right) = \prod_{c\in C_{census}} L\left(D_c|\pi_{D_c|\alpha}\right) = \prod_{c\in C_{census}} \left[ \frac{\left(\sum_i q_{c,\,i}\right)!}{q_{c,\,1}!q_{c,\,2}!\dots q_{c,\,M_{D_c}}!} \prod_i \pi_{c,\,i|\alpha}^{q_{c,\,i}} \right].$$

(11)

The estimated values of $\alpha$ were used to calculate census-equivalent population-level estimates including incidence rates, FoI, and susceptibility. Briefly, to obtain such an estimate on an arbitrary category $A$ that may stand for an estimated value or a binary variable that represents responses to a specific survey question, the group-level estimate of $A$ regarding the group of interest $G$ can be calculated as a weighted average:

$$\overline{A} = \frac{\sum_{i\in G} \alpha_i A_i}{\sum_{i\in G} \alpha_i}.$$

(12)

The estimated values of $\alpha$ were also used to obtain the census-equivalent population-level density by FoI and susceptibility level. When considering intervals $\boldsymbol{U} = \{U_1,\ U_2,\ \dots\}$ for $A$, the density in an interval $I_j$ can be calculated as

$$w_j = \frac{\sum_{i|A_i\in U_j} \alpha_i}{\sum_{i|A\in \boldsymbol{U}} \alpha_i}.$$

(13)

**2.3.4. Statistical inference of parameters.** We estimated $\Theta$ and $\alpha$ using the total likelihood $L = L_1\left(D^Q|\Theta\right) L_2\left(D|\alpha\right)$ of the Markov Chain Monte Carlo method together with the prior distributions and information as follows (see S2 Text for details):

- prior distributions for $\beta_{k\geq 2}$ were designed for the implementation of a Bayesian lasso framework [52]

• weak to non-informative priors were used for other parameters

• findings from published studies on vaccine effectiveness against JN.1 variants were also reflected [53–55]

For each of the four Markov Chain Monte Carlo chains, we generated 1,500 samples and discarded the first 500 warmup iterations to generate a total of 1,000 posterior samples. A total of 4,000 posterior samples were collected. We confirmed that R-hat statistics for all parameters were below 1.01 and that all default diagnostics of CmdStan returned no issues regarding convergence. The predictive performances for the binary outcome "whether or not infected with COVID-19 in February 2024" by sex in each decadal age group, and also by prefecture, were also evaluated, where the observed outcomes in all categories lay within the 95% Prediction Interval. (S2 Fig and S8 Data). All of the analyses were conducted in R (version 4.2.2) and Stan via CmdStan (version 2.34.0) [56–58]. (codes to reproduce the results are accessible at GitHub: https://github.com/pk2393/covid_qs_feb2024 and Zenodo: https://doi.org/10.5281/zenodo.17221098).

## 3. Results

### 3.1. Diagnosis-based Force of infection and relative hazard ratio by personal attribute at individual level

Table 1 summarizes the posterior estimates of $\beta_k$, the intercept and coefficients for covariates representing individual-level FoI as described in Eq. 3. The intercept $\beta_0$ represents the baseline national force of infection (FoI) among those aged 20–29 years without any additional effects related to personal attributes. An inverse relationship between the hazard ratio and age was observed. Among health condition factors, substantially high hazard ratios were observed in diabetes mellitus (1.8; 95% CrI 1.1, 2.9), neoplastic disorders (5.2; 95% CrI 3.1, 8.6), immunological suppression (2.6; 95% CrI 1.3, 4.6), respiratory

disease (2.2; 95% CrI 1.4, 3.3), and cardiovascular disease (2.3; 95% CrI 1.3, 3.9). A similar trend was suggested in liver disease. Obesity did not substantially modulate the relative hazard ratio (0.7; 95% CrI 0.3, 1.3). Additionally, smoking (1.3; 95% CrI 1.0, 1.8) and alcohol drinking (0.9; 95% CrI 0.6, 1.3) did not significantly alter the FoI. A previous history of infection (7.7; 95% CrI 5.6, 10.7) was strongly associated with higher FoI. Although not shown in Table 1, prefectural analysis revealed a mildly higher FoI in specific prefectures, including Ishikawa, Osaka, and Kagoshima (S3 Fig).

### 3.2. Protection against COVID-19 infection from infection or vaccination in February 2024

The protection or relative hazard reduction against COVID-19 from natural infection or vaccination at individual level is summarized in Fig 1. At the peak of protective immunity (i.e., at 30 days after the last immunizing event), protection levels provided by infections were higher overall than those by vaccines. At 30 days post-infection, the highest protection level (52.0% risk reduction; 95% CrI 33.2, 68.7) was against pre-XBB Omicron variants, followed by that against XBB.1.5 lineages (47.2%; 95% CrI 32.3, 60.7) and pre-Omicron variants (42.4%; 95% CrI 24.2, 60.5). At 30 days post-vaccination, the protection offered by the XBB.1.5 monovalent vaccine (45.1%; 95% CrI 37.8, 52.7) was the highest, followed by that of the Wuhan monovalent vaccine (36.3%; 95% CrI 19.8, 54.1) and the Wuhan-Omicron BA.1/5 bivalent vaccine (28.7%; 95% CrI 17.3, 40.6). The parameters underlying these estimates are provided in Table 2 and in S3 Table.

### 3.3. Statistical fitting of the respondent population to the census data

As described in the sections 3.1 and 3.2, we estimated the statistical weights for respondents that are needed to convert the result from the respondent group to that equivalent to the population in the 2020 census. Table 3 summarizes the

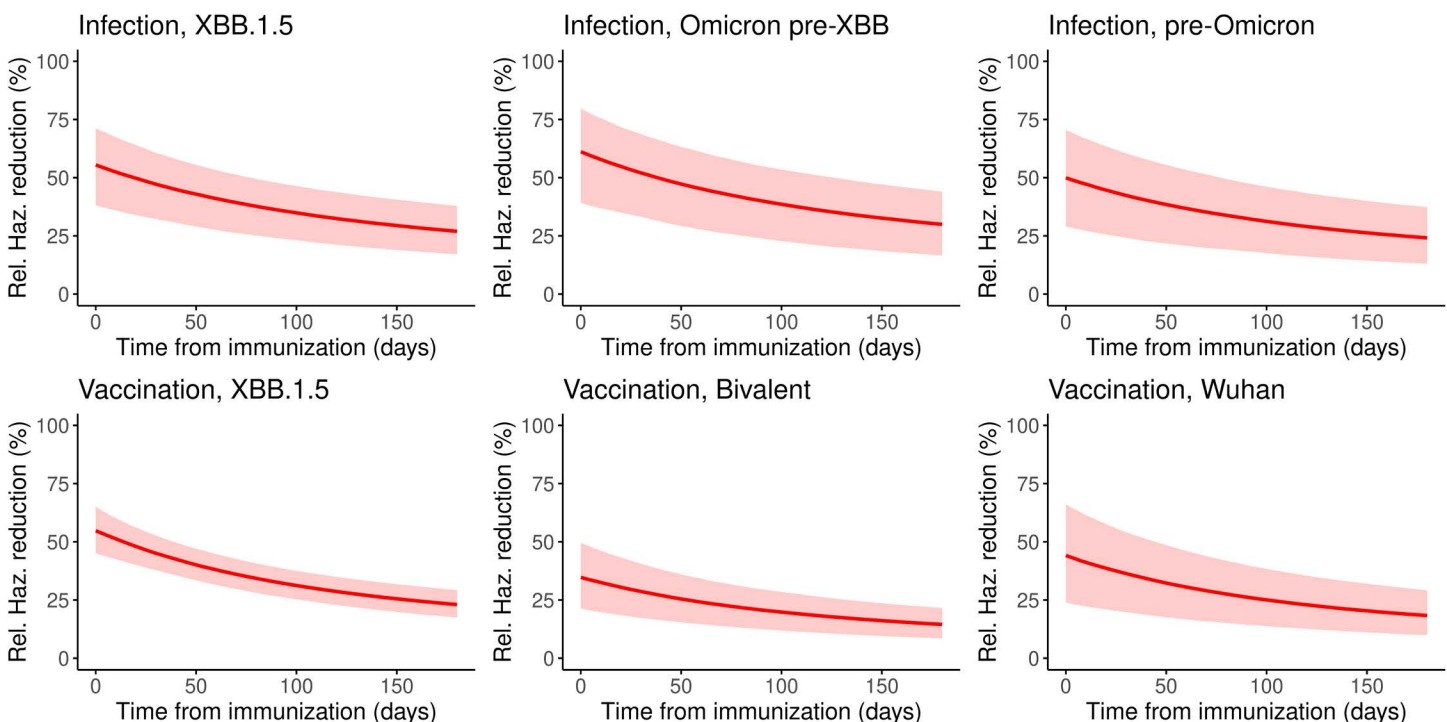

**Fig 1. Model estimates of protection levels and their decay over time by type of last immunizing event.** Each panel represents the decay of protective immunity (expressed as relative hazard reduction) by type of last immunizing event. In each panel, the horizontal axis shows the time since the last immunizing event.

**Table 2. Estimated peak protection (expressed as relative hazard reduction in percentage) at one month after the last immunizing event by type of vaccine or natural infection.**

| Last Immunizing Event | Estimated values (95% CrI) |
|---|---|
| Infection | |
| XBB sub lineages | 47.2 (32.3, 60.7) |
| pre-XBB Omicron | 52.0 (33.2, 68.7) |
| pre-Omircon | 42.4 (24.2, 60.5) |
| Vaccination | |
| XBB.1.5 | 45.1 (37.8, 52.7) |
| Wuhan+Omicron BA.1/5 | 28.7 (17.3, 40.6) |
| Wuhan | 36.3 (19.8, 54.1) |

**Table 3. Estimated sex ratio and proportion of population by age group.**

| Category | Estimated values (95% CrI) | Census (95% CI)* |
|---|---|---|
| Male, national | 49.37 (49.36, 49.38) | 49.41 (49.40, 49.42) |
| Male by age group (years) | | |
| 20-29 | 13.43 (13.42, 13.44) | 13.50 (13.49, 13.51) |
| 30-39 | 15.52 (15.51, 15.53) | 15.50 (15.48, 15.51) |
| 40-49 | 20.09 (20.07, 20.10) | 20.06 (20.04, 20.08) |
| 50-59 | 18.04 (18.03, 18.05) | 18.04 (18.03, 18.06) |
| 60-69 | 16.61 (16.60, 16.62) | 16.63 (16.61, 16.64) |
| 70 and over | 16.31 (16.30, 16.32) | 16.27 (16.26, 16.28) |
| Female by age group (years) | | |
| 20-29 | 12.72 (12.71, 12.73) | 12.71 (12.70, 12.72) |
| 30-39 | 14.73 (14.72, 14.74) | 14.72 (14.71, 14.74) |
| 40-49 | 19.24 (19.23, 19.25) | 19.24 (19.22, 19.26) |
| 50-59 | 17.67 (17.66, 17.68) | 17.67 (17.66, 17.69) |
| 60-69 | 17.03 (17.02, 17.04) | 17.03 (17.02, 17.05) |
| 70 and over | 18.62 (18.60, 18.63) | 18.62 (18.60, 18.63) |

* The confidence interval was calculated using the Goodman method [59].

following statistics, which were estimated as the weighted sum of answers from respondents: 1) proportions of men and women in the national adult population, 2) proportions by age group in the national male population, and 3) proportions by age group in the national female population.

The proportion of men was estimated at 49.37% (95% CrI 49.36, 49.38), compared with 49.41% based on the 2020 census. Among the male population, the estimated proportions of the population for the 20–29, 30–39, 40–49, 50–59, 60–69, and 70-year-and-older age groups were 13.43% (95% CrI 13.42, 13.44), 15.52% (95% CrI 15.51, 15.53), 20.09% (95% CrI 20.07, 20.10), 18.04% (95% CrI 18.03, 18.05), 16.61% (95% CrI 16.60, 16.62), and 16.31% (95% CrI 16.30, 16.32), respectively. Except for the slight discrepancies in the 20–29-year and 70-year-and-older age groups, these results mostly aligned with the 2020 census. The estimated age-specific proportions of the female population for the same age groups were 12.72% (95% CrI 12.71, 12.73), 14.73% (95% CrI 14.72, 14.74), 19.24% (95% CrI 19.23, 19.25), 17.67% (95% CrI 17.66, 17.68), 17.03% (95% CrI 17.02, 17.04), and 18.62% (95% CrI 18.60, 18.63), respectively; these results were in line with the 2020 census. The estimated population distribution by prefecture and job type distribution by age group are in S9-S12 Data. Despite slight discrepancies between the 95% CrI and the census data (especially in data

categories with sparse absolute counts in the respondent group), the weighted demographics closely followed the pattern in the census data.

### 3.4. COVID-19 incidence by age group in february 2024

By applying the estimated statistical weights for each respondent, we estimated the population-level incidence rate of COVID-19 during February 1st – 29th, 2024, at the national level in Japan. Fig 2 shows the estimated COVID-19 incidence rates in the 20–29, 30–39, 40–49, 50–59, 60–69, and 70-year-and-older age groups. The highest estimated incidence was 4.73% (95% CrI; 4.17, 5.38) in those aged 20–29 years, whereas the lowest estimated incidence was 1.19% (95% CrI; 0.94, 1.47) in the 60–69-year age group. Overall, the incidence rates were high in the younger population and low in the older population, with the exception of a higher incidence in the 70-year-and-older age group compared with the 60–69-year age group.

For each age group, the blue point represents the median posterior incidence rate estimate, and the error bar represents the 95% credible interval.

The results of the comparison between external data and our estimates are provided in the S4 Table, S4 and S5 Figs, and S13 Data. Briefly, the comparison with the national case counts provided by Moderna (only point estimates provided) revealed that for age group 20–59 and age 60 or over, our estimates were higher by 70.0% (95% CrI: 58.7, 82.2) and 75.7% (50.9, 102), respectively. (S4 Table) Comparisons with the serological survey from residual serum and donor blood are described in S4 and S5 Figs and S13 Data. Overall, for all age groups, the rates with infection history (positive

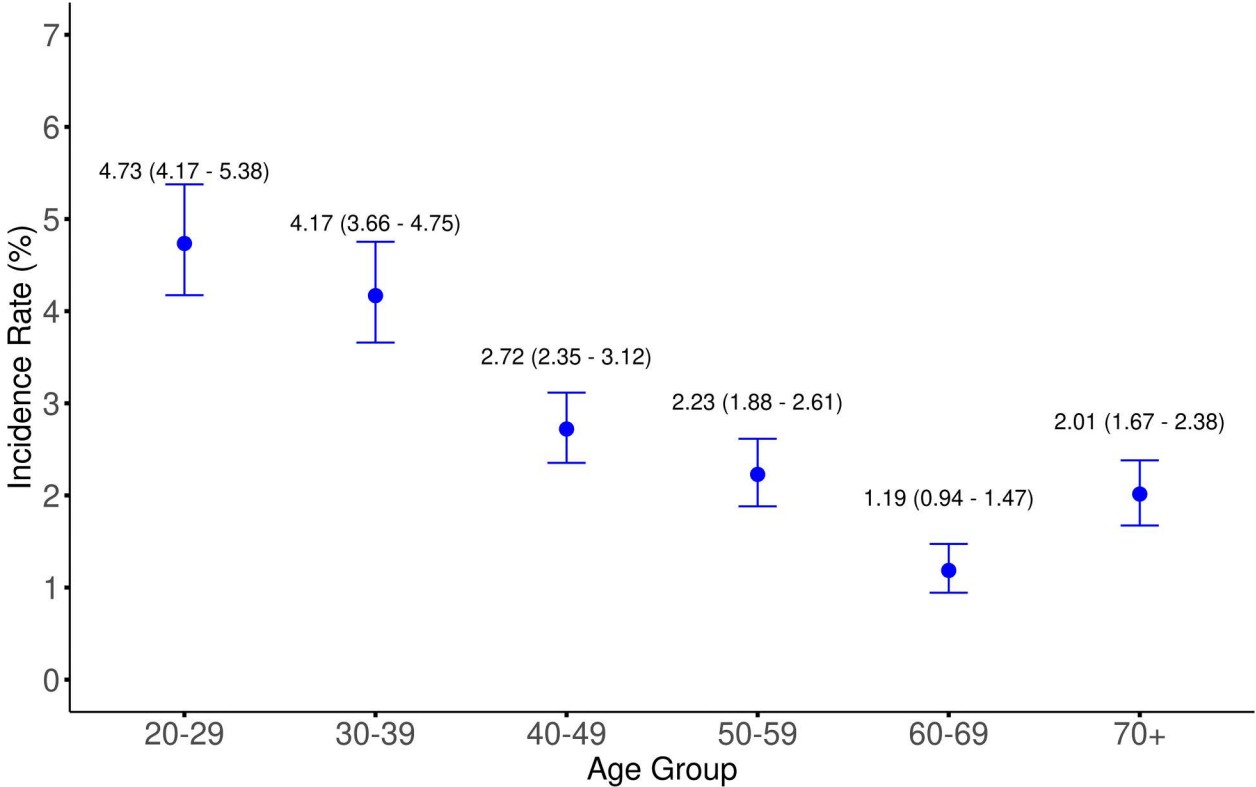

**Fig 2. Estimated national incidence rates of COVID-19 by age group in February 2024.**

anti-N-antibody) up to January and March 2024 by our estimates were mostly lower by 20–40% for those younger than 60, but these gaps were slightly smaller for those aged 70 and older for both residual serum and donor blood samples. For "any exposure" (positive anti-S-antibody), the discrepancy was smaller, where in residual serum the positivity rate was almost >98% regardless of the sex or age group, whereas our estimates were mostly lower by 10–20% overall compared with residual blood samples.

### 3.5. Mapping the diagnosis-based force of infection and immune protection by age group in february 2024

Fig 3 illustrates the distribution of FoI by each age group from Feb 1st to Feb 29th, 2024, which was estimated from the questionnaire survey data collected during Mar 7th to Mar 13th, 2024. The distributions were created by adapting the statistical weights of respondents by adjusting the questionnaire-based estimates to those equivalent to the national population (See Materials and Methods and S2 Text for further details on the statistical weights). Among those aged 20–29 years, a mild bimodal pattern in the FoI distribution was observed, with over 50% of the population concentrated in the low FoI range (< 0.02) and a substantial proportion also observed in the high FoI range (approximately 0.06–0.14). A very small fraction with FoI > 0.2 was also observed as a slight upward tick in the right tail of the distribution. In older age groups, the distribution was more heavily concentrated in the low FoI range. The bimodal pattern weakened in older ages and was visually unidentifiable among those aged 70 years and older.

About the characteristic bimodal pattern in those aged 20–29 and those aged 30–39, we further compared the underlying attributes (including covariates in the above-mentioned model) between those in the "lower-FoI (FoI<0.045)" mode and "higher-FoI (FoI>=0.045)" mode. (The threshold 0.045 was decided by visual inspection of Fig 3) As shown in S5 Table, weighted estimates of these attributes revealed that the rates of previous vaccination, past infection, and several

## Force of Infection by Age Group

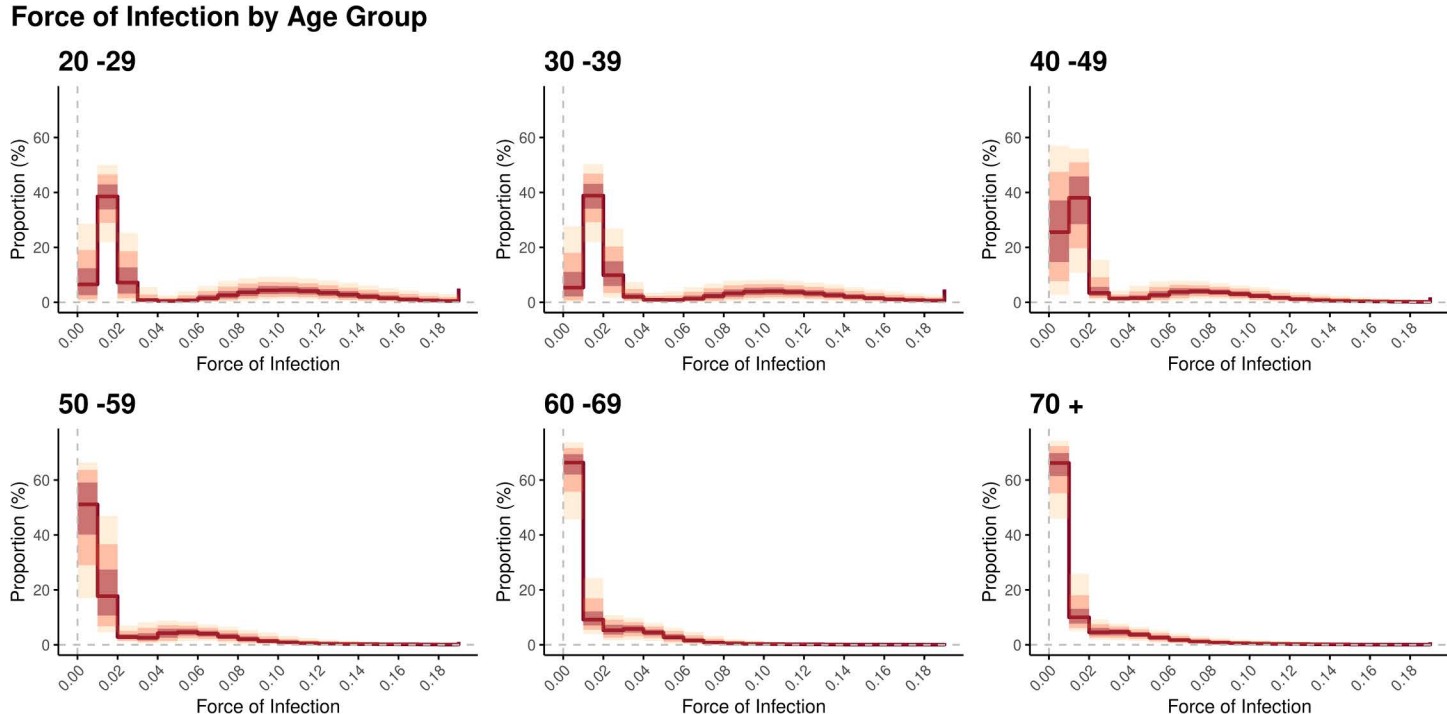

**Fig 3. Estimated distribution of the diagnosis-based force of infection in Japan by age group in February 2024.** Each panel shows the fraction of the age-stratified population that lies in discretized intervals of the force of infection (expressed with the unit of per month). In each panel, the red solid line represents the median in each interval, and the red, pink, and yellow bands represent the 50%, 80%, and 95% credible intervals, respectively.

comorbidities were higher in the higher-FoI group, whereas the rate of "never-infected" was higher in the lower-FoI group, despite the lower vaccination rate in this group.

Fig 4 illustrates the distribution of individual protection as relative hazard reduction in each age group. The statistical weights of respondents were adapted in the same manner as in Fig 3. Most of the individuals aged 20–29 years were concentrated in the low protection zone (i.e., below 30%), and a monotonic decline toward the higher protection levels was observed. This pattern weakened in older age groups in which increasingly bimodal distributions of protection levels were observed.

The findings in Figs 3 and 4 are illustrated as two-dimensional heatmaps in Fig 5, in which the colors in each cell represent the value of the posterior average. One key finding is the high-density area in the low protection-low FoI region in all age groups. As shown in S6 Fig, further inspection with stratification by the last immunizing event revealed that those with no exposure contributed to the low (no) protection-low FoI region, and the remaining density in the low protection- low FoI region was almost solely attributable to those whose last exposure was vaccine, regardless of the age group. In addition to the concentration in the low protection–low FoI region, a substantial fraction in the relatively high FoI range that was centered at roughly 0.10 was observed in younger age groups. This pattern was less evident in the groups aged 60–69 years and 70 years and older. The fraction of those with FoI > 0.2 also tended to decrease in older age groups.

## 4. Discussion

We proposed a statistical framework based on a questionnaire survey via the Internet to estimate the monthly population-wide incidence of COVID-19, the effect of biological and social immune correlates, and the risk heterogeneity of individuals to COVID-19 based on personal attributes. Importantly, we observed the inverse relationship between

**Immune Protection by Age Group**

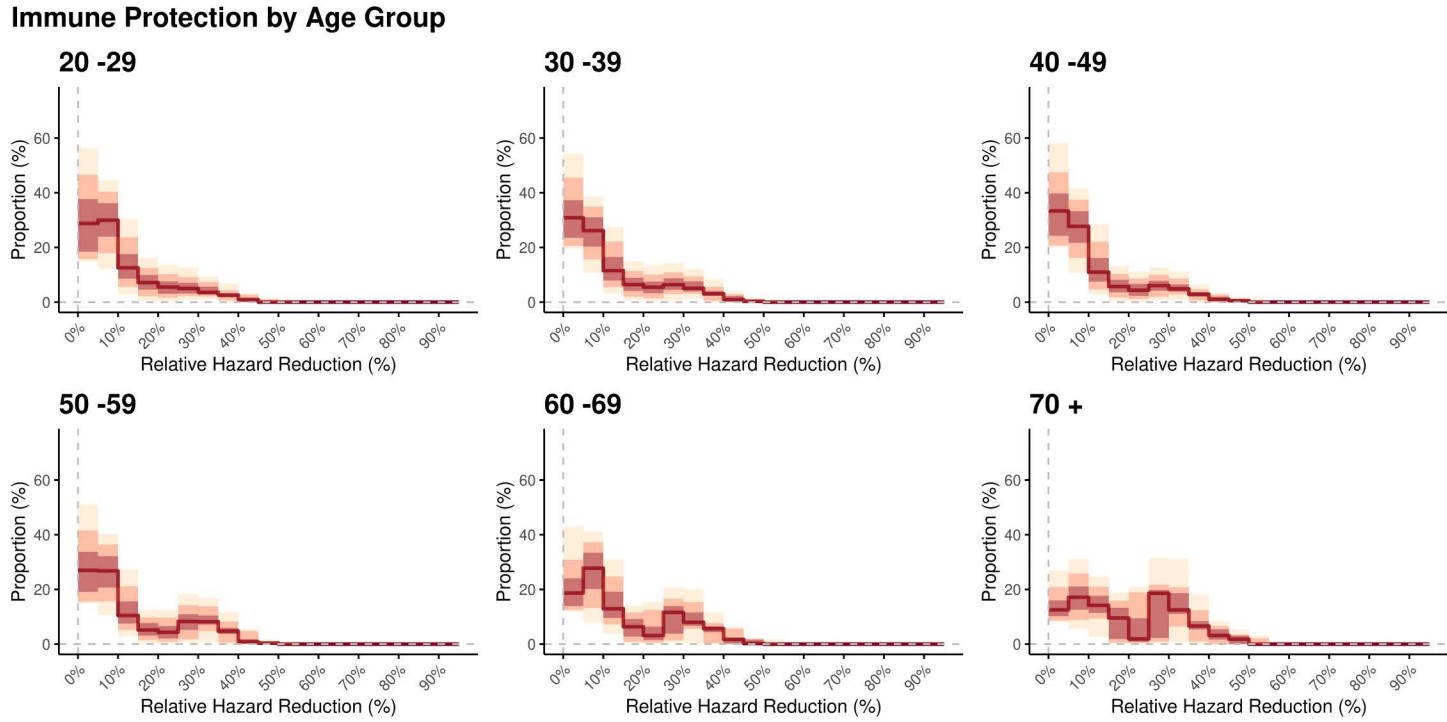

**Fig 4. Estimated distribution of national immune protection (expressed as relative hazard reduction) by age group in February 2024.** Each panel shows the fraction of the age-stratified population that lies in discretized intervals of the protection level. In each panel, the red solid line represents the median in each interval, and the red, pink, and yellow bands represent the 50%, 80%, and 95% credible intervals, respectively.

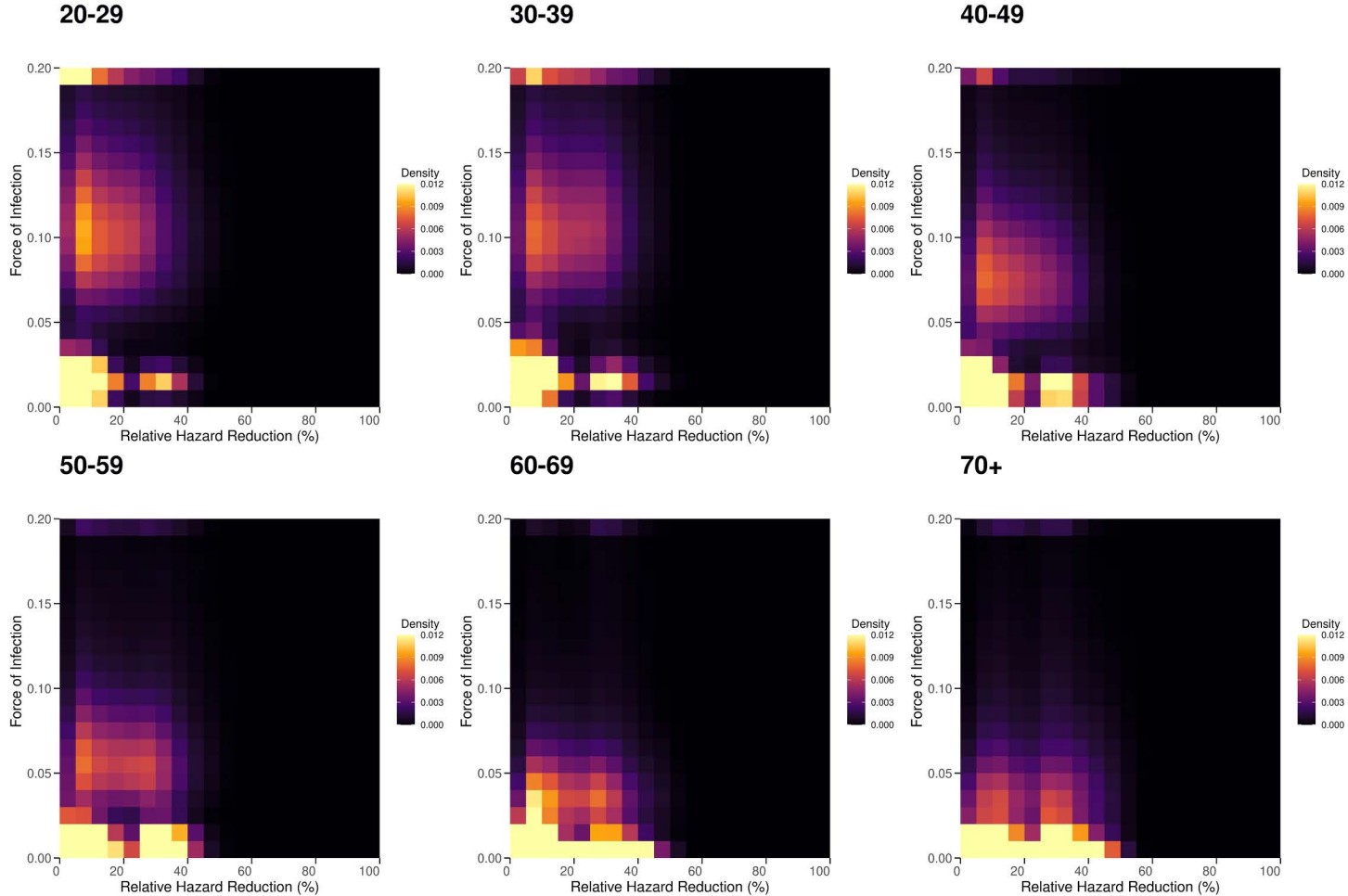

**Fig 5. Heatmaps of the distribution of the diagnosis-based force of infection and relative hazard reduction of infection by age group.** In each panel, the cells represent the population fractions within discretized two-dimensional intervals of the force of infection and protection level (expressed as the relative hazard reduction). The cells with colors closer to yellow represent high density, whereas those closer to black indicate low density.

age and FoI after adjustment for the protection offered by acquired immunity and substantial heterogeneity within each age group, as highlighted in the heatmaps in Fig 5. Furthermore, we revealed that personal attributes, including health conditions, are important modulators of the diagnosis-based FoI despite most individuals having received vaccination or experienced natural infection at least once.

Several key take-home messages emerged from this study. First, the relatively low protection against JN.1 variants appeared to be induced by vaccination and natural infection. This effect was especially prominent following vaccination with the bivalent Wuhan + Omicron BA. 1/5 vaccine. This result confirms the validity of published studies on JN.1 variants in Japan and aligns with the findings of neutralization studies that suggest substantial immune evasion of JN.1 from immunity provided by exposure to pre-JN.1 infection or vaccines [53,54,60–63]. Revealing the mechanism underlying the especially low protection conferred by the bivalent Wuhan + Omicron BA. 1/5 vaccine is beyond the scope of our study; however, our result may be an epidemiological presentation of "immune imprinting" [64–66].

Second, the effects of personal attributes on the risk of infection were also notable and revealed that various underlying health conditions linked with impaired immunity or respiratory or cardiovascular dysfunctions were risk factors for

contracting COVID-19. These factors have already been identified as determinants of severe manifestations or death associated with COVID-19 [34–38]. Our finding suggests that, in addition to the importance of vaccinations to curb the risk of developing severe illness by COVID-19, the significance of preventive measures such as wearing masks or avoiding crowded spaces cannot be ignored in individuals with underlying health conditions. In addition to health-related risk, the substantially elevated risk of infection in those with a history of previous infection may reflect unidentified risk factors, including behavioral aspects (e.g., those at risk of death may seek medical attention more often than others) or socioeconomic status, which were not explicitly captured in this study.

Third, another important finding from a population-wide viewpoint is the heterogenous distribution of FoI and protection levels, as highlighted in Figs 3, 4, 5 and S6. A larger proportion of the younger population was observed in high FoI–low protection regions. By contrast, an opposite pattern was observed in the older population, which was mostly concentrated in low FoI regions with mild heterogeneity, potentially reflecting the recency of infection or vaccination. It was also revealed that those with no exposure concentrate in the low protection -low (no) FoI region across age groups. This may reflect more cautious contact patterns among some unvaccinated individuals. Because of the difficulty capturing this type of heterogeneity from routine epidemiological surveillance regardless of the disease of interest, our finding is especially important and may guide the effective and efficient implementation of epidemic countermeasures [23]. Furthermore, as post-acute sequelae of COVID-19 gain increasingly more attention, our finding highlights the need to promote vaccination, especially in the working-age population [8,9,67].

Two technical advantages of our framework were identified. First, our findings are based exclusively on a rapidly implementable and simple Internet-based approach that reveals the characteristics of personal immune protection. This point is critical because individual-level profiles regarding immunity against COVID-19 are growing increasingly complex given repeated vaccination and infections and newly emerging SARS-CoV-2 variants. In this context, our framework is expected to be an important addition to an array of conventional approaches that evaluate the effectiveness of vaccine- or infection-induced immunity based on prospective recruitment of study participants over several months or retrospective analysis [21,22,68–70].

Second, our proposed approach of linking the questionnaire respondents to several categories of the census enabled the estimation of both incidence rates and immune protection at the population level. Using our framework to estimate incidence rates may be a complementary approach to epidemiological surveillance, which is essential for tracking key epidemiological indicators, including case-fatality and case-hospitalization ratios. Such an approach is especially useful given that conventional epidemiological surveillance is increasingly limited. Population-level immune protection cannot be estimated with conventional approaches given current data limitations [24,25]. Thus, our framework may be a useful and cost-effective alternative to achieve this aim. Moreover, by widening its scope to infectious diseases other than COVID-19, our framework may guide the implementation of more effective and efficient public health interventions based on risk stratification by age group [23].

Several key limitations of our study should be discussed. First, due to the nature of Internet-based questionnaire surveys, potential biases may arise from several factors such as the selection of respondents, the correctness of self-reported infection histories (that are not based on medical tests or records), the variability in healthcare- or test-seeking behavior among respondents that can vary by the milder symptoms and reduced awareness of COVID-19 infection due to vaccination [71,72], and the presence of close exposure to COVID-19 cases or local epidemic intensity. About the bias inherent to respondents, it is possible that statistical matching of respondents to the 2020 census may have not fully accounted for the bias in respondent as discussed later. About the reliability of self-reported infection histories, comparisons between our estimates and seroepidemiological surveys suggested some discrepancies, especially regarding the lower estimates of the rates with personal history of infection (corresponding to anti-nucleocapsid antibody positivity in seroepidemiological surveys) in our study (S4 and S5 Figs) [45–48]. In contrast, the comparison between national incidence from our estimates and the Moderna estimate during February 2024 suggests a higher incidence in our

study. Though the provided estimates in seroepidemiological surveys and the Moderna website are not directly comparable with each other, these gaps may have arisen for various reasons that can be explained by methodological differences, including incorrect answers from the respondents and ascertainment bias, or the difference in the target population. About the difference in target population including individual-level information on health conditions or personal attributes, it is difficult to explore further because of the lack of such information for both seroepidemiological surveys and Moderna website; however, developing a approach to fill this gap is an important future scope of study for evaluating the robustness of our estimation on the effect of individual-level risk factors.

Second key limitation is that, small discrepancies between the 2020 census and our estimates of population distribution by sex, age, prefecture, and job category may be attributable to the small sample size of respondents representing each stratum in the 2020 census, and may have slightly distorted our population-level estimates of incidence rates or protection levels. Our future research scope includes enhancing the robustness of population-level estimates by conducting larger-scale surveys, potentially with the engagement of several survey companies. Another technical limitation is that, because complete, timestamped infection and vaccination histories were not collected in the present study, survival or time-to-event analyses were not feasible. Future surveys that record events at least the preceding several months (including non-infection intervals) could enable further analyses. Finally, our usage of a discrete monthly time scale in the questionnaire survey and the categorization of vaccines and SARS-CoV-2 variants may have been an oversimplification.

In conclusion, we estimated the incidence rate, immune protection of the population, and relative hazard of infection attributable to individual factors and converted these estimates to population-level results using our statistical weighting method. Our framework is an important addition to existing epidemiological surveillance programs and studies that focus on vaccine- or infection-induced immunity. Implementing this framework not only would provide a better understanding of epidemiological dynamics but also may enhance our knowledge of how immunological and non-immunological individual-level attributes affect the risk of infection across many infectious diseases and not only COVID-19.

## Supporting information

**S1 Text. Survey items and the questionnaire used in this study.**
(PDF)

**S2 Text. Prior distributions and information used for Bayesian inference by Markov Chain Monte Carlo method.**
(PDF)

**S1 Table. Descriptive analysis of survey respondents.**
(PDF)

**S2 Table. Number of respondents by job category linked to the Census.**
(PDF)

**S3 Table. Estimated values of parameters characterizing immune protection dynamics.**
(PDF)

**S4 Table. Comparison of our national incidence estimates with the case count estimates provided by the Moderna Inc. website (URL: https://moderna-epi-report.jp/) in February 2024. [26,27].**
(PDF)

**S5 Table. Comparison of the proportion of key personal-level attributes between those in the "lower-FoI (FoI<0.045)" mode and "higher-FoI(FoI>=0.045)" for those aged 20–29, weighted results as national estimates.**
(PDF)

**S1 Fig. Distribution of the time since last immunizing event stratified by last exposure type and the presence of infection in February 2024.** Each column represents the time distribution by last exposure type, which is stratified by infection event in February 2024. (Upper row: "Positive (Feb 2024)", Lower row: "Negative (Feb 2024)").
(TIF)

**S2 Fig. Posterior Predictive Check for COVID-19 infection in Feb 2024.** In each panel, labels on top represent the corresponding age group and sex. The predicted outcomes from the posterior distribution are shown in sky-blue histograms with dashed vertical lines representing the 95% Prediction Interval. The red solid vertical line in each panel shows the observed number of infections in our questionnaire survey.
(TIF)

**S3 Fig. Posterior estimates of the prefectural effects that modify personal force of infection.** The posterior distribution of the random effect for each prefecture is shown as a violin plot. The dot indicates the posterior median, and the solid vertical line represents the 95% Credible Interval.
(TIF)

**S4 Fig. Comparison of weighted estimates of past exposure to COVID-19 infection and vaccination with serological surveys (donated blood [47,48] and the residual of clinical serum samples at commercial clinical testing laboratories [45,46]) in the Japanese male population in January and March 2024.** For serological surveys, results of anti-nucleocapsid antibody positivity are shown in the "Past Infection" panels, whereas results of anti-spike antibody positivity are shown in the "Any Exposure" panels. *results shown for "70+-year" age group for the residual serum sample survey are actually for the "70-79" age group in the survey because of the difference in age group stratification. [45,46]. **no results for the "70+-year" age group and "anti-spike antibody" in the donated blood survey. [47,48].
(TIF)

**S5 Fig. Comparison of weighted estimates on the past exposure to COVID-19 infection and vaccination with serological surveys (donated blood [47,48] and the residual of clinical serum samples at commercial clinical testing laboratories [45,46]) in the Japanese female population in January and March 2024.** For serological surveys, results of anti-nucleocapsid antibody positivity are shown in the "Past Infection" panels, whereas results of anti-spike antibody positivity are shown in the "Any Exposure" panels. *results shown for "70+-year" age group for the residual serum sample survey are actually from the "70-79-year" age group in the survey because of the difference in age group stratification. [45,46]. **no results for the "70+-year" age group and "anti-spike antibody" in the donated blood survey. [47,48].
(TIF)

**S6 Fig. Heatmaps of the distribution of the diagnosis-based force of infection and relative hazard reduction of infection by age group and by the type of last immunizing event.** The left, middle, and right columns represent panels for "None (never-immunized)", "Infection (last event = infection)", and "Vaccine (last event = vaccination)", respectively. In each panel, the cells represent the population fractions within discretized two-dimensional intervals of the force of infection and protection level (expressed as the relative hazard reduction). The cells with colors closer to yellow represent high density, whereas those closer to black indicate low density.
(TIF)

**S1 Data. Questionnaire survey results.**
(CSV)

**S2 Data. Census data of the male and female population by age group.**
(CSV)

**S3 Data. Census data of the male population by age group and prefecture.**
(CSV)

**S4 Data. Census data of the female population by age group and prefecture.**
(CSV)

**S5 Data. Census data of the male population by job category and by age group.**
(CSV)

**S6 Data. Census data of the female population by job category and by age group.**
(CSV)

**S7 Data. Case count estimates in February 2024 retrieved from the Moderna Inc. website (URL: https://moder-na-epi-report.jp/).** [26,27].
(CSV)

**S8 Data. Posterior predictive check by prefecture.**
(CSV)

**S9 Data. Estimated prefecture-wise population distribution by age group, male population.**
(CSV)

**S10 Data. Estimated prefecture-wise population distribution by age group, female population.**
(CSV)

**S11 Data. Estimated job type distribution by age group, male population.**
(CSV)

**S12 Data. Estimated job type distribution by age group, female population.**
(CSV)

**S13 Data. Serological survey estimates and our posterior estimates for anti-N and anti-S antibody positivity rates.**
(CSV)

## Acknowledgments

We thank Anahid Pinchis, MBA, BSc from Edanz (https://jp.edanz.com/ac) for editing a draft of this manuscript.

## Author contributions

**Conceptualization:** Yuta Okada, Hiroshi Nishiura.

**Data curation:** Yuta Okada.

**Formal analysis:** Yuta Okada.

**Investigation:** Yuta Okada, Hiroshi Nishiura.

**Methodology:** Yuta Okada, Hiroshi Nishiura.

**Project administration:** Hiroshi Nishiura.

**Supervision:** Hiroshi Nishiura.

**Validation:** Yuta Okada.

**Visualization:** Yuta Okada, Hiroshi Nishiura.

**Writing – original draft:** Yuta Okada, Hiroshi Nishiura.

**Writing – review & editing:** Yuta Okada, Hiroshi Nishiura.

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
