## [Decision Letter · Decision Letter 0]

7 Jul 2025

Reconstructing the force of infection and immune fraction of the population via a single snapshot survey: a case study of COVID-19 in Japan

PLOS Computational Biology

Dear Dr. Nishiura,

Thank you for submitting your manuscript to PLOS Computational Biology. After careful consideration, we feel that it has merit but does not fully meet PLOS Computational Biology's publication criteria as it currently stands. Therefore, we invite you to submit a revised version of the manuscript that addresses the points raised during the review process.

Please submit your revised manuscript within 60 days Sep 06 2025 11:59PM. If you will need more time than this to complete your revisions, please reply to this message or contact the journal office at ploscompbiol@plos.org. Please include the following items when submitting your revised manuscript:

We look forward to receiving your revised manuscript.

Kind regards,

Tom Britton

Academic Editor

PLOS Computational Biology

Jennifer Flegg

Section Editor

PLOS Computational Biology

**Additional Editor Comments :**

Academic editor

First, excuse me for the long delay for the revision, but it was unusually hard to find referees: I had to ask 20 (!) candidates to get two who accepted ...

Now the paper has been read by two experts in the field and, more briefly, by myself. We all find it interesting, but also lacking some important features in order to be accepted in PLoS Comp Bio. The first referee asks for a better explanation of the methodology, the second referee raises several points to be better addressed. Myself, I lack a bit of focus on computational aspects, given that is the main focus of the journal. Please address all pointas raised by the referees before reasubmitting.

Kind regards, Tom Britton

**Journal Requirements:**

1) Thank you for stating that "All datasets are shared as the supplementary material, and moreover, available from the author's github." Please update your Data Availability Statement in the online submission form to include direct links to github datasets.

2) Your current Financial Disclosure states several funds. However, your funding information on the submission form indicates one fund. Please ensure that the funders and grant numbers match between the Financial Disclosure field and the Funding Information tab in your submission form. Note that the funders must be provided in the same order in both places as well.

**Reviewers' comments:**

Reviewer's Responses to Questions

Reviewer #1: In this paper, the authors report estimates of COVID-19 case incidence, strain-specific susceptibility, and vaccine effectiveness from a snapshot online questionnaire in Japan. The work yields interesting epidemiological results and proposes a novel methodological approach. I can see many people in public health being interested in this work as an inexpensive way to generate insights that have until now only been possible with very expensive studies.

I did find the innovation in the paper difficult to access. There were very few details of the survey questions, and the key methodological innovation was not described early on in natural language. I think PLOS Computational Biology still allows a "methods first" structure. If so, I'd suggest adding a concise natural language methods section to explain what was asked in the questionnaire and how the Bayesian framework works (not a load of equations—natural language about what the equations are doing). The current methods with equations could then be included in the supplementary material.

I think this is very good and important work, but it was hard to tell from a single careful reading, and I feel very familiar with the questions and the methods.

Detailed comments

Abstract – should give a hint of how this can be achieved without any biological outcome data.

Results – it's still very much a mystery at this point how we can infer these quantities from a single snapshot survey. A "results first" ordering might not make sense for this work.

Page 13, line +3 (pdf numbering): need to define and motivate the β (beta) parameter. Very abrupt for the reader as written.

Page 16, +2: the reweighting seems to be more of a methods detail, or perhaps the first results?

Page 18, -4: what time period do the results relate to? How wide was the snapshot?

Page 24, line 1: These findings should be presented as part of the results, not only highlighted from the discussion. How big were the differences? Are they material to the other claims in the paper?

Page 24, -8: no details at this point about the monthly time scales.

Page 29, +3: vaccine effectiveness against self-reported infection? The details of the outcome measure need to be stated more clearly, especially given that the questionnaire was in Japanese and the article is in English. What were participants actually asked? The details of the questionnaire are not clear. Given recent LLM advances in translation, perhaps the authors could provide the original Japanese text and one or two LLM-generated translations to give readers a better chance to understand the outcome.

Page 30, +1: This appears to be the central methodological innovation. The questionnaire is designed to inform a multi-group attack-rate calculation. So people's risk of infection in a similar group in a similar place should be consistent with the rest of their group. This does make sense – can we know how good a fit the data are to the model? This structure should be adding considerably more constraint than a simple regression approach—so does it look like it's working?

Reviewer #2: This article investigates the estimation of COVID-19 incidence and protection conferred by previous infections and vaccinations using a single cross-sectional Internet-based survey conducted in Japan in February 2024. The authors propose a statistical framework to reconstruct the force of infection (FoI) and individual-level protection based on reported immunization history and health-related covariates. While the study is methodologically interesting, I have major concerns regarding several points:

1/ The study would benefit significantly from validation against objective external data sources, such as case counts, serological survey, hospitalisation data. While the authors mention some comparison with sero-surveys in the supplementary materials, this is limited and not well integrated into the main results. Such validation is crucial for confirming the accuracy of incidence and protection estimates derived from self-reported data.

2/ It is unclear what it means to model and report protection at one-month post-immunization (Figure 1), when many immunizing events—particularly natural infections or vaccinations—occurred well before the study period, even more than two years before for the pre-Omicron infections. Although the protection curve is modeled (or rather, depicted) over six months post-event, the actual data used for fitting includes participants with much older immunization histories. This creates a disconnect between the temporal assumptions of the model and the actual time lags in the data, raising concerns about too much extrapolation. A clarification would be welcome.

3/Related to the previous point I think the authors should provide either a figure or table showing raw or minimally processed data, such as:

- Type of last immunizing event (infection or vaccination),

- Time since the event,

- Whether the respondent was infected in February 2024.

4/ Many individuals report last immunization months (or years) prior to the survey, yet no reinfection was observed between that time and February 2024. Would it be beneficial to the study to use a survival analysis or time-to-event model here? This non-infection interval contains valuable information that is currently unused, such as the protection given by immunization not only at the time of the survey but also during all opportunities of infections in the meantime.

5/ The bimodal distribution of FoI in younger individuals (Figure 3) is interesting but should be explored more thoroughly. The authors should clarify what immunization types or demographics are associated with each mode.

6/ In relation to Figure 5, the authors mention a subgroup in the “low force of infection – low protection” region. If possible, could the authors clarify whether these individuals are a specific group of unexposed (e.g., never infected, low contact) or whether their presence is simply a statistical expectation of the model?

**Have the authors made all data and (if applicable) computational code underlying the findings in their manuscript fully available?**

Reviewer #1: **No:** Not sure if the code were made available. Sorry if I missed that. I saw the data statement which seemed reasonable.

Reviewer #2: Yes

PLOS authors have the option to publish the peer review history of their article (what does this mean? ). If published, this will include your full peer review and any attached files.

**Do you want your identity to be public for this peer review?** For information about this choice, including consent withdrawal, please see our Privacy Policy .

Reviewer #1: No

Reviewer #2: No

**Figure resubmission:**

**Reproducibility:**



---

## [Decision Letter · Decision Letter 1]

31 Dec 2025

PCOMPBIOL-D-25-00610R1

Reconstructing the force of infection and immune fraction of the population via a single snapshot survey: a case study of COVID-19 in Japan

PLOS Computational Biology

Dear Dr. Nishiura,

Thank you for submitting your manuscript to PLOS Computational Biology. After careful consideration, we feel that it has merit but does not fully meet PLOS Computational Biology's publication criteria as it currently stands. Therefore, we invite you to submit a revised version of the manuscript that addresses the points raised during the review process.

We look forward to receiving your revised manuscript.

Kind regards,

Tom Britton

Academic Editor

PLOS Computational Biology

Jennifer Flegg

Section Editor

PLOS Computational Biology

**Additional Editor Comments:**

Academic editor

The revised version has been read by one of the reviewers and myself. Unfortunately we have not been able to reach reviewer 1, so I have checked their comments and your responses (below). Reviewer 2 is not fully satisfied with the rev ised version. I encourage you to address their comments with great detail in the next revision. If not I will not recommend the manuscript for publication.

As for the responses to comments of referee 1 it looks fine as far as I can judge. The only question is in your comments 8) and 10) where you write "We should have ...". I assume you mean that "We have ..."?

Kind regards, Tom Britton

**Journal Requirements:**

1) Please ensure that the funders and grant numbers match between the Financial Disclosure field and the Funding Information tab in your submission form. Note that the funders must be provided in the same order in both places as well.

**Reviewers' comments:**

Reviewer's Responses to Questions

**Comments to the Authors:**

Reviewer #2: I appreciated the efforts of the authors in responding to the questions of the reviewers, but I still have concerns regarding some results and the claims made in the study.

For instance, In really appreciated that they included external studies. However think more efforts should be made to discuss the discrepancy between their study and the external study (Moderna). If available, I think this paper would benefit from comparing the risk factors from those other studies. If seroprevalence estimates are different, are the trends in risk factors still reliable ?

I’m not convinced that increasing the size of the study as it is proposed by the authors in line 518-519 would benefit so much for survey-based estimate of the inference. Given the design of the study, the biases are essentially similar to that of case-based estimates of the incidence. I suggest the authors outline some potential sources of bias of their study. The first that come to my mind are asymptomatic infections that may lead to underestimating the incidence, and the second is a high incidence in the household, family members, colleagues that may increase the respondent probability of reporting symptoms, and third, high incidence within the population.

There are other issues with the asymptomatic infections. Vaccination could induce more asymptomatic and paucisymptomatic infections (see for instance Joseph et al., JAMA, 2022 and many other studies). Those could be detected with serological surveillance but the proposed approach would fail to identify them. This doesn’t mean the approach is not useful, as it may still be able to identify and quantify risk factors for severe infections. In this sense it is an interesting approach, that can be implemented quickly. However, I suggest that the authors tone down their claim that they can assess the force of infection as they claim in the title. The authors themselves outline this as a limitation on line 150 where they say that they detect « symptomatic COVID-19 infections ».

**Have the authors made all data and (if applicable) computational code underlying the findings in their manuscript fully available?**

Reviewer #2: Yes

PLOS authors have the option to publish the peer review history of their article (what does this mean? ). If published, this will include your full peer review and any attached files.

**Do you want your identity to be public for this peer review?** For information about this choice, including consent withdrawal, please see our Privacy Policy .

Reviewer #2: No

**Figure resubmission:**

**Reproducibility:**



---

## [Decision Letter · Decision Letter 2]

5 Feb 2026

Dear Dr. Nishiura,

We are pleased to inform you that your manuscript 'Reconstructing the incidence rate and immune fraction of the population via a single snapshot survey: a case study of COVID-19 in Japan' has been provisionally accepted for publication in PLOS Computational Biology.

Best regards,

Tom Britton

Academic Editor

PLOS Computational Biology

Jennifer Flegg

Section Editor

PLOS Computational Biology

Academic editor

Reviewer 2 is now also happy with the manuscript why I suggest that the paper is accepted for publication.

Kind regards, Tom Britton

Reviewer's Responses to Questions

**Comments to the Authors:**

Reviewer #2: The authors responded adequately to the reviews. I believe the paper is now suitable for publication.

**Have the authors made all data and (if applicable) computational code underlying the findings in their manuscript fully available?**

Reviewer #2: Yes

PLOS authors have the option to publish the peer review history of their article (what does this mean? ). If published, this will include your full peer review and any attached files.

**Do you want your identity to be public for this peer review?** For information about this choice, including consent withdrawal, please see our Privacy Policy .

Reviewer #2: No

---

## [Editor Report · Acceptance letter]

PCOMPBIOL-D-25-00610R2

Reconstructing the incidence rate and immune fraction of the population via a single snapshot survey: a case study of COVID-19 in Japan

Dear Dr Nishiura,

I am pleased to inform you that your manuscript has been formally accepted for publication in PLOS Computational Biology. Your manuscript is now with our production department and you will be notified of the publication date in due course.

With kind regards,

Judit Kozma
